# A Combination of Melatonin and Ethanol Treatment Improves Postharvest Quality in Bitter Melon Fruit

**DOI:** 10.3390/foods9101376

**Published:** 2020-09-27

**Authors:** Xiaoxue Lin, Li Wang, Yuanyuan Hou, Yonghua Zheng, Peng Jin

**Affiliations:** College of Food Science and Technology, Nanjing Agricultural University, Nanjing 210095, China; 2018108026@njau.edu.cn (X.L.); 2017208028@njau.edu.cn (L.W.); 2019208027@njau.edu.cn (Y.H.); zhengyh@njau.edu.cn (Y.Z.)

**Keywords:** bitter melon, postharvest quality, melatonin, ethanol, response surface methodology

## Abstract

Central composite design (CCD), utilized with three independent variables, verified that the optimal treatment conditions in bitter melon fruit were melatonin (MT) concentration of 120 µmol L^−1^, ethanol concentration of 6%, and immersing time of 10 min. Under optimal conditions, the experimental values of firmness, chilling injury (CI) index, and weight loss were shown as 27.81 N, 65.625%, and 0.815%, respectively. Moreover, the combined effect of MT and ethanol on CI and physiological quality in postharvest bitter melon fruit stored at 4 °C was investigated. It was found that the combined treatment contributed to the reduced CI symptoms and inhibited ion leakage and malondialdehyde (MDA) accumulation. Moreover, higher levels of chlorophyll, total soluble solids (TSSs), soluble sugar, soluble protein, and ascorbic acid (AsA) were observed in comparison with the control group. Furthermore, the synthesis of total phenols and flavonoids in bitter melon was greatly promoted. Therefore, the combination of MT and ethanol could have the potential for alleviating CI and maintaining postharvest quality for the duration of cold storage.

## 1. Introduction

Bitter melon (*Momordica charantia* L.) is a cucurbitaceous plant that grows in tropical, subtropical, and temperate regions. It is very susceptible, and the loss of fruit quality is shown as enlarged seeds, soft tissue, yellow pericarp, and loss of bitterness. According to Han et al. [1], 1-MCP was effective in inhibiting ethylene production, maintaining postharvest quality and enhancing the capacity of antioxidant activity. In another study, Dong et al. [2] confirmed that bitter melon treated with 42 °C for 5 min can maintain the best physiological quality without obvious chilling injury (CI). Low temperature is a practical and convenient method for extending storage time by slowing down metabolic activities in bitter melon. However, the effect is often limited by the rise in CI symptoms, which included dark watery patches and black surface discoloration. Thus, it is vital to develop a safe and effective postharvest treatment to alleviate low-temperature damage and maintain the quality of bitter melon fruit.

Melatonin (*N*-acetyl-5-methoxytryptamine, MT), a kind of bioactive substance synthesized with tryptophan as a matrix, was first discovered in the pineal gland [3]. Previous studies have demonstrated that exogenous MT is a beneficial hormone for preserving the quality of sweet cherries [4] and prolonging the fruit senescence of peach fruit [5]. Recently, Liu et al. [6] stated that MT may increase the cold resistance of litchi fruit via suppressing the enhancements of ion leakage and malondialdehyde (MDA) and regulating the metabolism of energy and proline. In addition, CI of MT-treated pomegranate fruit was greatly inhibited by decreasing ion leakage and MDA content [7]. Other studies have depicted that the quality maintenance of horticultural crops subjected to MT may be ascribed to the higher levels of chlorophyll, total phenols, and anthocyanin [8,9]. However, little information concerning the effect of MT on CI control and quality maintenance of bitter melon has been reported.

Ethanol is a kind of natural volatile compound with the characteristics of low toxicity and environmental friendliness that has been generally recognized as a safe (GRAS) substance by the Food and Drug Administration (FDA) [10]. Previous studies have shown that ethanol has the potential for delaying the ripening of “Royal Gala” apples [11] and retaining high marketability of edible petunia flowers and cherry tomatoes [12,13]. As a well-known preservative, ethanol has played an essential role in horticultural products during the postharvest phase. Moreover, as an organic solvent, ethanol can promote the dissolution of water-insoluble substances such as MT. Thus, the combined application of MT and ethanol can play a positive role in fruit preservation. 

MT and ethanol treatment alone can both reduce CI of postharvest fruits and vegetables. However, there are few reports on the combined application of MT and ethanol to alleviate CI of fruit. Studies have shown that combined treatments can help to increase the resistance of peach fruit to cold stress [14,15] and the quality maintenance of broccoli florets [16]. Moreover, response surface methodology (RSM) is a statistical method for solving multiple variable problems. The basic principle of RSM is to describe the model equations, define the influence of test variables on the response, and determine correlations between test variables and responses [17]. It is effective for the prediction of optimal performance conditions of multiple variables with a minimum number of experiments. Central composite design (CCD) of RSM can provide a reasonable amount of information for lack-of-fit testing without involving an abnormally large amount of design points [18]. It has been used for the optimization of postharvest apple [19] and pear [20] preservation. Therefore, in the present study, the optimal conditions of a combined treatment of MT and ethanol were evaluated by RSM. Furthermore, the effect of the combined treatment on the quality of bitter melon fruit was studied. It may provide a reference for the application of combined treatments in the cold tolerance enhancement of bitter melon.

## 2. Materials and Methods 

### 2.1. Plant Material

The bitter melons (*Momordica charantia* L. cv. “Green champion”) for research were hand-harvested in Shandong, China at commercial maturity with full fruit. Then, fruit with uniform size and color, which also had an absence of malformation, over-ripeness, and damage, were selected.

### 2.2. Single-Factor Experiment of MT Treatment

As MT was light-sensitive, the experiment was carried out under dark conditions. Moreover, MT is insoluble in water, and 0.0625% ethanol was used as a cosolvent to promote the dissolution of MT. In order to determine the appropriate concentration range of MT in the following RSM experiment, the selected bitter melon fruit were soaked in the 50, 100, 150, and 200 µmol L^−1^ MT solution for 10 min at ambient temperature. Fruit dipped in distilled water were set as the control. All the fruit were dried naturally and separately packed in 0.01 mm polyethylene plastic bags with 4 fruit in each bag, and the opening of the bags was loosely wrapped with two ordinary rubber bands. Each treatment was replicated three times with 40 fruit, and the experiment was conducted twice. The chilling injury (CI) index of bitter melon fruit in each group was recorded on day 20 at 4 ± 1 °C (95% relative humidity), following a shelf life at 25 °C for three days.

### 2.3. Single-Factor Experiment of Ethanol Treatment

To determine the appropriate concentration range of ethanol in the RSM experiment, the selected bitter melon fruit were soaked in the 3%, 6%, 9%, 12%, and 15% ethanol solution (prepared with ethanol) for 10 min at ambient temperature. The fruit dipped in distilled water were set as the control. Each treatment was replicated three times with 40 fruit, and the experiment was conducted twice. CI index was recorded on day 20 at 4 ± 1 °C (95% relative humidity), following a shelf life at 25 °C for three days.

### 2.4. RSM Design

For the purpose of monitoring the effects of treatment conditions on quality changes of bitter melon, a three-factor CCD with 20 runs was implemented and is shown in Table 1. It was also used to identify the relationship between independent and response variables and achieve the optimized treatment conditions. Runs in the design involved eight factorial points, six axial points (the distance between the two axis points of each independent variable and the center point was 1.68), and six center points. Runs at the central point were duplicated to estimate the pure error of the experimental design, and the lack-of-fit of the experimental model could be obtained. The concentration of MT (x_1_, 50–150 µmol L^−1^), the concentration of ethanol (x_2_, 2.5–9.5%), and soaking time (x_3_, 5–15 min) were selected as process design variables based on the results of single-factor experiments. Meanwhile, firmness (Y_1_), CI index (Y_2_), and weight loss (Y_3_) were chosen as the response variables. The results of the last day were recorded. The influence of unexplained variability results from external factors was minimized for the randomness of the experiment. 

The response data of three factors to three dependent variables were investigated, and the quadratic polynomial fitting was also carried out. The equation of the prediction model is reported in Equation (1):(1)$$Y_{i}=\omega_{0}+\sum_{i=1}^{3}\omega_{i}X_{i}+\sum_{i=1}^{3}\omega_{ii}X_{i}{}^{2}+\sum_{i\neq j=1}^{3}\omega_{ij}X_{i}X_{j}$$
where Y_i_ was the representation of the predicted response function (firmness, CI index, and weight loss) and X_i_ and X_j_ were the independent variables; ω_0_, ω_i_, ω_ii_ and ω_ij_ were the constant, linear, quadratic, and interaction coefficients, respectively.

### 2.5. MT-Ethanol Combined Treatment

In order to investigate the effect of MT and ethanol on the quality maintenance in bitter melon, the fruit were immersed for 10 min in MT—ethanol solution and distilled water was used as control. The treatment condition was derived from the result of the RSM experiment. Each treatment had 150 fruit for three replications, and the experiment was conducted twice. At 4-day intervals for the duration, the fruit were sampled, and the frozen tissues were stored at −20 °C prior to analysis. Finally, an additional three days of storage at 25 °C after each sampling date was required to record the CI index.

### 2.6. Weight Loss, CI Index, and Firmness Assessments

Weight loss was presented as the percentage weight loss in comparison with the initial weight, which was determined by weighing bitter melon fruit instantly after air-drying and at every sampling date.

The CI index of bitter melon was determined according to the method of Zhao et al. [21]. The fruit were classified as 0 to 4 on the CI scale, where 0 = no chilling, 1 = mild injury, 2 = moderate (marketability-limited), 3 = slightly severe with moderate decay, and 4 = exceedingly severe with enormously secondary infections. The CI index was expressed as the following formula (Equation (2)):(2)$$\mathrm{CI Index}\;(\%)\;=\;\frac{\sum \left(\mathrm{CI scale}\;\times \;\mathrm{the number of fruit in each scale}\right)}{\left(\;4\;\times \;\mathrm{the total number of fruits}\right)}\;\times \;100$$

The firmness of bitter melon fruit after peeling in the two opposite positions of the equator and after internal seed removal was measured by a TA-XT2i texture analyzer (TMS-Pro, Food Technology Corporation, Sterling, VA, USA). The diameter probe (6 mm) was pushed 18 mm into the flesh at a speed of 10 mm s^−1^. Eight replicate fruit were selected and performed in triplicate. The maximum penetration force was regarded as the firmness, and the unit of Newtons (N) was taken to manifest the results.

### 2.7. Determinations of Fruit Ion Leakage and MDA Content

Ion leakage was determined according to Zhang et al. [22] with some modifications. Briefly, 18 discs (6 mm diameter) were excised from the equator part of six fruit and put into three graduated tubes equally. The conductivity of tissue (C_0_) was measured with a conductivity meter immediately after being shaken. Half an hour later, the solution (C_1_) was measured following treatment by boiling for 10 min and then measured again after cooling as C_2_. The formula is as follows (Equation (3)):(3)$$\mathrm{Ion Leakage}\;(\%)\;=\;\left(\;C_{2}\;-C_{0}\;\right)/\left(\;C_{1}\;-C_{0}\;\right)\;\times 100$$

MDA content was determined by the thiobarbituric acid method [23]. Fruit tissue (1.0 g) was homogenized in the icy water bath with 5.0 mL of 100 g L^−1^ trichloroacetic acid (TCA) solution and then centrifuged. Absorbance was measured at 600, 532, and 450 nm, and the results were expressed as g kg^−1^ of bitter melon fresh weight basis.

### 2.8. Ascorbic acid (AsA), TSS, and Chlorophyll Content Assays

The AsA content was expressed on a fresh weight basis as g kg^−1^ and determined as the assay described by Kampfenkel et al. [24], with some modifications. Fruit tissue (1.0 g) was homogenized in the icy water bath with 5.0 mL of 5% TCA solution and then centrifuged. A 5.0 mL of reaction system was applied, in which 1.0 mL extract, 1.0 mL 5 % TCA, 1.0 mL absolute ethanol, 0.5 mL 0.4% phosphoric acid, 1.0 mL 0.5% phenanthroline, and 0.5 mL 0.03% ferric chloride (FeCl_3_) were contained. The mixture was incubated in the 30 °C water bath for 1 h after vortex mixing, and then the absorbance of the red chelate produced after the reaction was recorded at 534 nm. 

To determine the total soluble solid (TSS) content, flesh tissue in the equatorial region of eight fruit were chopped and wrapped in three layers of gauze, followed by a hand squeezing. The extracted juice was placed in a portable refractometer (PAL-1, Atago, Tokyo, Japan) and analyzed in triplicate. Values were read and displayed as a percentage.

Chlorophyll content was determined in accordance with the method of Li et al. [8]. Fruit chlorophyll (2.0 g) was extracted from the fresh peels with pure acetone. Absorbance was measured at 663 and 645 nm, and the results were expressed as g kg^−1^ of bitter melon fresh weight basis.

### 2.9. Measurements of Total Phenols, Total Flavonoids, Soluble Sugar, and Soluble Protein Content

The contents of total phenols and total flavonoids were determined as the assay described by Liu et al. [25], with some modifications, and expressed as g kg^−1^. Fruit tissue (1.0 g) was homogenized in the icy water bath with 5.0 mL of extract (1% of hydrochloric acid and 70% of ethanol) and then incubated in the 60 °C water bath for 1 h with shaking every 10 min. After cooling, the homogenate was centrifuged. For the total phenols assay, a 2.0-mL of the reaction system was applied, in which 150 μL extract, 50 μL distilled water, 1.0 mL Folin–Ciocalteau reagent, and 0.8 mL 75 g L^−1^ sodium carbonate (Na_2_CO_3_) solution were contained. The mixture was incubated in a 30 °C water bath for 1 h after vortex mixing, and then the absorbance was recorded at 765 nm. 

For the total flavonoids assay, an extraction solution (1.5 mL) mixed with 1.5 mL aluminum chloride (AlCl_3_) solution and 2.0 mL 0.1 mol L^−1^ sodium acetate solution (pH 5.5) was incubated in water at a constant temperature of 30 °C for 15 min. Finally, the absorbance of the solvent was measured at 430 nm, and the results of total phenols and total flavonoids were both expressed as g kg^−1^ of bitter melon fresh weight basis.

For assaying of the content of soluble sugar, 1.0 g of bitter melon tissue was ground with 5.0 mL of distilled water. The homogenate was bathed in water at 80 °C for 1 h at first, and then treated with ultrasound for 5 min under 35 °C; finally, it was centrifuged at 13,000× *g* for 20 min at 4 °C. The extraction solution (80 μL), mixed with 1920 μL distilled water, 1.0 mL 5% phenol, and 5.0 mL of concentrated sulfuric acid (H_2_SO_4_), was incubated in water at a constant temperature of 40 °C for 20 min. Finally, the absorbance of the solvent was measured at 490 nm with a spectrophotometer, and the result was expressed as g of glucose per kg of bitter melon fresh weight basis. Moreover, sucrose was used as a standard in this measurement for quantitative analysis.

Soluble protein estimation was carried out by consulting the method of Liu et al. [26]. A 6.0-mL reaction system was applied, in which 1.0 mL of extract and 5.0 mL of comassie brilliant blue G-250 solution were contained. Bovine serum albumin was used as the standard product, and the result was expressed as g kg^−1^ of bitter melon fresh weight basis.

### 2.10. Statistical Analysis

Experiments were carried out in a completely randomized design, and the results were described as the mean ± standard deviation (SD). Statistical analysis was performed by using Origin 8.5 software for drafting, SAS 9.2 software for one-way variance analysis (ANOVA, Duncan’s multiple range tests, *p* < 0.05), and Design-Expert 8.0.6 (trial version, Stat Ease Inc., Minneapolis, MN, USA) for CCD.

## 3. Results

### 3.1. Single-Factor Experiments

Compared with the control group, a significant decrease of approximately 33.3% (*p* < 0.05) in the CI index was noted for 100 µmol L^−1^ MT-treated fruit on day 20, and 100 µmol L^−1^ MT also presented the lowest CI index among all of these treatments (Table 2). In addition, the CI index of bitter melon treated with 150 and 200 µmol L^−1^ MT both recorded no significant difference (*p* > 0.05) compared with the control fruit. Therefore, 100 µmol L^−1^ of MT was adopted as the optimum concentration.

No significant difference (*p* > 0.05) of CI index was noted between 6% and 9% ethanol-treated groups on day 20 (Table 2). The CI index of these two groups also had no significant differences compared with the control on day 20 (Table 2). The CI index in bitter melon fruit with 6% ethanol treatment was 42.9% lower than that in control, while no significant differences (*p* > 0.05) among several other concentrations were noted (Table 2). Thus, the level of 6% was the best choice for further use.

### 3.2. Model Fitting and Statistical Analysis of CCD 

According to the results of single-factor experiments, the optimum treatment conditions were further simulated by the CCD project to obtain maximum firmness, minimum CI index, and minimum weight loss of the bitter melon fruit. The response values corresponding to 20 groups are presented in Table 3. The correlation of the response data between three factors and three dependent variables are expressed by quadratic multiple regression equations as follows (Equations (4)–(6)):Y_1_ = 25.56 + 1.38x_1_ − 1.40x_2_ + 0.60x_3_ + 0.24x_1_x_2_ + 1.38x_1_x_3_ + 2.56x_2_x_3_ + 0.35x_1_^2^ − 2.08x_2_^2^ − 2.06x_3_^2^(4)
Y_2_ = 40.4 − 6.06x_1_ + 6.06x_2_ + 3.42x_3_ + 14.47x_1_x_2_ + 5.84x_1_x_3_ − 0.41x_2_x_3_ + 17.46x_1_^2^ + 8.57x_2_^2^ + 11.93x_3_^2^(5)
Y_3_ = 0.77 − 0.17x_1_ − 0.4x_2_ − 0.25x_3_ − 0.16x_1_x_2_ + 0.52x_1_x_3_ − 0.35x_2_x_3_ + 0.69x_1_^2^ + 0.37x_2_^2^ + 0.42x_3_^2^(6)

The coefficients of regression equations of RSM for firmness, CI, and weight loss models were analyzed by ANOVA (Table 4). The *p*-values of the three models were obviously either less than 0.0001 or equal to 0.0001, which revealed that the firmness, CI, and weight loss models were all remarkably significant (*p* < 0.01). Moreover, determination coefficients (*R*^2^) of the predicted models of firmness, CI index, and weight loss were 0.9939, 0.9270, and 0.9689, respectively. It indicated that accurate predictions of response behaviors were reached with the proposed experimental models. Besides, the lack-of-fit of the three models was statistically significant (*p* < 0.05), illustrating the high degree of reliability and repeatability of the models.

The linear coefficients (x_1_ and x_3_), interaction (x_1_x_3_ and x_2_x_3_), and quadratic (x_1_^2^) were represented in Table 4, which had a significantly positive effect on firmness (*p* < 0.01). However, the ethanol concentration (x_2_) and the quadratic values (x_2_^2^ and x_3_^2^) exhibited a significantly negative effect on firmness (*p* < 0.01). 

The ANOVA of the CI index showed that MT concentration (x_1_) and ethanol concentration (x_2_) both had a significant effect on the model (*p* < 0.05). Moreover, all of the three quadratic term coefficients and cross-product coefficients (x_1_x_2_) were also remarkably significant on the CI index due to the *p*-value <0.01. 

The influences of other coefficient items on the model of weight loss were significant except for the cross-product coefficients of x_1_x_2_ (*p* < 0.05), especially the items of x_2_, x_1_x_3_, x_1_^2^, and x_3_^2^, whose *p*-values were below 0.0001.

### 3.3. Response Surface Analysis of CCD

Three-dimensional (3D) response surface data graphs were displayed in Figure 1. The interaction effects of the three test factors on three dependent variables were obviously visualized, with each two of the independent factors affecting the response value and the other factor remaining constant. The influences of independent variables on response values were observed.

A dramatic effect of the three factors on the firmness model of bitter melon was clearly observed. The level of firmness was the highest when immersion time was between 9 and 10 min and ethanol concentration was fixed at level zero. Moreover, the surface of the firmness model became steeper with an increase of ethanol concentration and treatment time, which demonstrated a remarkably significant interaction effect on firmness between the two factors (*p* < 0.0001). In addition, MT concentration and ethanol concentration appeared to have an insignificant interaction on the firmness of the surface (*p* > 0.05).

The 3D graphs of the CI index showed that simultaneously reduced concentrations of MT and ethanol increased the slope of the curved surface along with a significant interaction (*p* < 0.01) in the 10 min immersion. Another two gentle influence surfaces, determined by x_1_x_3_ and x_2_x_3_, were found, which revealed an insignificant interaction effect of x_1_x_3_ and x_2_x_3_ on the model of the CI index (*p* > 0.05).

With respect to the 3D plots of weight loss, 100–125 µmol L^−1^ MT was the most effective treatment when time was fixed at level zero. The 3D response surface of x_1_x_2_ was the mildest in comparison with other surfaces of the weight loss model, indicating that the interaction of x_1_x_2_ was not significant (*p* > 0.05). In addition, an increased weight loss was obtained with the long immersion time and reduced MT concentration in the meantime, when the ethanol level was fixed at 6%. Keeping the MT concentration at an intermediate level, the weight loss of fruit experienced an increasing trend with the increase of ethanol levels, which proved that extended ethanol concentration negatively affected the weight loss of bitter melon.

### 3.4. Validation Test of CCD

The optimized treatment conditions for firmness, CI index, and weight loss by RSM were MT concentration (116.11 µmol L^−1^), ethanol concentration (5.54%), and immersing time (10.03 min). Taking the convenience of operation into account, the concentration of 120 µmol L^−1^ and 5.5% and the time of 10 min were chosen. Then, the verification experiment (Table 5) was carried out in triplicate by using the optimized conditions from the model simulation to evaluate and confirm the validity of the model. The results of validation tests showed that three treatments could significantly (*p* < 0.05) reduce the weight loss and CI of bitter melon fruit and maintain a higher level of firmness. The MT–ethanol combined treatment was the most effective compared with the other groups.

### 3.5. Effect of MT–Ethanol Combined Treatment on CI Index, Ion Leakage, and MDA Content

The CI index and ion leakage exhibited an increasing trend during the storage period (Figure 2A,B). The combined treatment started to show CI symptoms on day 8 of the storage duration, whereas the control showed symptoms earlier, on day 4. In addition, as observed in Figure 2B, ion leakage of the MT- and ethanol-treated group did not increase rapidly until day 12, compared to the control, with a sharp increase from day 4. Ion leakage in the treated fruit was reduced to a large extent in comparison with the control. The content of MDA in the combined treatment group changed slowly during the whole storage, while it increased and accumulated rapidly in the control group after day 12. The MDA level was significantly decreased by 48.8% under the combined treatment of MT and ethanol.

### 3.6. Effect of MT–Ethanol Combined Treatment on Chlorophyll Content, Soluble Solids, and Vitamin C Content

The trend of ascorbic acid showed a similar pattern to total soluble solids, while some differences can be distinctly noticed from Figure 3A. In addition, the ascorbic acid content of control fruit was a little higher than that of MT–ethanol-treated ones before 4 d of storage, and then it declined in a very quick manner.

The TSS content of fruit tissue (Figure 3B) exhibited an increasing trend initially and then kept reducing in the remaining storage period. The climacteric time for the treated fruit was on day 12. At the end of storage, the TSS content in the bitter melon fruit with a combined treatment of MT and ethanol was 5.9% higher than that in control.

An obvious decline of chlorophyll was noticed during the whole storage (Figure 3C); however, MT and ethanol treatment effectively maintained the chlorophyll content at a high level of 0.1 g kg^−1^ on day 20, which was statistically significant (*p* < 0.05).

### 3.7. Effect of MT–Ethanol Combined Treatment on Total Phenols, Total Flavonoids, Soluble Sugar, and Soluble Protein Content

The combined treatment of MT and ethanol effectively promoted the synthesis and accumulation of total phenols and flavonoids in bitter melon fruit. The total phenol content of the treated group was significantly higher than that of control (*p* < 0.05) during the whole storage period. The content of total flavonoids in the combined treatment accumulated slowly in the early stage of storage, while it increased rapidly from day 16 and reached 0.2 g kg^−1^ at the end of storage.

Bitter melon treated with MT in combination with ethanol demonstrated excellent higher levels of total soluble sugar and soluble protein with storage duration compared with the control; the explicit results are shown in Figure 4C,D. During the storage period, soluble sugar content on the treated and control groups declined 33.8% and 43.8%, and a significant difference existed between days 12 and 20 (*p* < 0.05). The largest amount of soluble protein in treated samples was recorded on day 16, which was 9.0% higher than control. The MT combined with ethanol treatment effectively retarded the reduction of soluble protein, which was important for the bitter melon to maintain its quality for the duration of senescence.

## 4. Discussion

In the pre-experiment of MT concentration, the treatment of 100 µmol L^−1^ MT was the best for increasing the resistance of bitter melon to cold stress. Similar results were reported in the research of Cao et al. [27] in peach fruit. Jannatizadeh et al. [28] reported that the cold tolerance enhancement of MT may involve the regulations of enzyme activities and gene expressions related to energy metabolism and membrane lipid metabolism. When MT concentration increased to 200 µmol L^−1^ or ethanol concentration increased to 12% and 15%, serious CI symptoms would appear. It is probable that higher levels of MT or ethanol may cause chemical injury to the cells of bitter melon fruit. Meanwhile, the osmotic stress caused by high osmotic pressure may arise, which will result in a negative effect of the MT and ethanol treatment. Moreover, the 3D graph of the CI index determined by MT and ethanol concentration confirmed that a concentration below a certain range could not produce a remarkable effect on alleviating CI of bitter melon fruit. All of the 3D plots of the response surface experiment in the present study reached an optimal value in the experimental limit, which indicated appropriate independent variable ranges.

Ion leakage, which is a consequence of cell membrane damage, is an essential parameter to evaluate the degree of CI encountered in postharvest fruit [21]. MDA can aggravate membrane injury; consequently, the degree of membrane lipid peroxidation can be indirectly obtained by measuring the content of MDA [29]. Jin et al. [30] found that the altering of membrane integrity and occurring of phospholipid peroxidation were due to the accumulation of reactive oxygen species (ROS) to a certain extent. The result in the current research showed that the MT–ethanol treatment led to a lower ion leakage and MDA content compared with the control, which may be associated with the avoidance of ROS accumulation. A previous study had reported that the accumulation of reactive oxygen may be induced by membrane integrity loss and membrane lipid peroxidation [23]. Shao et al. [31] found that the pericarp browning of wampee fruit treated with ethanol was effectively restrained, which may be related to the cell membrane protection caused by the ethanol-induced decrease in MDA content. A previous study concluded that the application of MT promoted the increase of cold resistance of *Camellia sinensis* L. by increasing the activity of antioxidant enzymes and upregulating the transcript abundance of antioxidant genes [8]. Jannatizadeh et al. [7] reported that the cold stress of pomegranate fruit treated with MT was greatly slowed down, which was associated with the regulation of ROS and membrane lipid metabolism. Moreover, MT itself can scavenge ROS, mainly by providing electrons, and it made itself an indole cation with low toxicity. Meanwhile, the nonenzymatic antioxidant system further scavenged ROS. Then, MT transformed into *N*1-acetyl-*N*2-formyl-5-methoxycanine amide (AFMK), which had a stronger antioxidative effect than MT [32]. Consequently, MT and ethanol inhibited the increase of ion leakage and MDA accumulation via scavenging ROS directly and indirectly, which greatly reduced the damage of cell membrane integrity caused by the low temperature. Thus, MT-ethanol treatment can alleviate CI of bitter melon fruit. However, further study of the combined treatment on physiological changes should be investigated.

A growing number of studies have shown that MT and ethanol alone or in combination with other treatments often illustrate a dramatic effect on maintaining the quality and reducing CI of cold-sensitive fruit during storage. It has been reported that ethanol combined with lower relative humidity conditions played an active role in maintaining the postharvest quality of apple, while ethanol alone was not efficient in postharvest preservation [11]. Hu et al. [3] reported that exogenous MT treatment maintained an acceptable overall quality of cassava fruit via reducing the degradation of ascorbic acid and starch during storage. In the present study, the combined treatment of MT and ethanol had remarkable effects on chlorophyll, soluble sugar, and ascorbic acid retention. Various metabolic activities are still in progress in the postharvest entity of fruits and vegetables, then a large amount of soluble sugar demanded to be decomposed for energy supply. The decomposition of chlorophyll and ascorbic acid is the manifestation of postharvest fruit senescence [33]. In addition, the increase in soluble protein before 16 d could have a connection with the production of cold-resistant protein induced by ethanol. A previous study also reported that the enhancement of postharvest fruit ripening and quality mediated by MT might be the result of the protein alteration related to the ripening process [34]. As a summer season vegetable, bitter melon continues to ripen after harvest and produces high amounts of ethylene; therefore, a climacteric of TSS can be noted in our present data. The physiological activities of the postharvest fruit yielded a reducing trend of soluble sugar. Moreover, the comprehensive effect of high molecular carbohydrate decomposition and respiratory consumption made TSSs have a trend of rising initially and then declining. Phenols and flavonoids in fruit are important antioxidants and play an important role in the response to environmental stress [35]. It is also crucial for the nutritional quality maintenance of fruit. As shown in Figure 4A,B, the accumulation of total phenols and flavonoids was significantly (*p* < 0.05) promoted by the combined treatment of MT and ethanol. As reported by Liu et al. [25], the antioxidant capacity of strawberry was greatly promoted by MT via increasing the accumulation of total phenols, flavonoids, and endogenous MT. Shao et al. [31] stated that ethanol treatment can effectively maintain fruit quality and increase antioxidant capacity by altering the related enzyme activities of phenol metabolism. Therefore, the combined treatment of MT and ethanol can maintain the postharvest quality of bitter melon fruit by keeping higher levels of chlorophyll, TSSs, soluble sugar, soluble protein, and AsA. Meanwhile, the combined treatment can also promote the synthesis of total phenols and flavonoids during cold storage at 4 °C.

## 5. Conclusions

The optimal conditions obtained by using the design of RSM were as follows: MT concentration of 120 µmol L^−1^, ethanol concentration of 6%, and immersing time of 10 min. Then, the effect of the combined treatment based on the optimal conditions of bitter melon fruit during cold storage was studied. Data showed that the MT–ethanol treatment can effectively alleviate CI and membrane lipid peroxidation of bitter melon. Moreover, the combined treatment of MT and ethanol can delay the decline of fruit quality by maintaining the nutrient content at a high level. More experiments are scheduled to explore the mechanism of the MT–ethanol treatment on cold resistance and the enhancement and quality maintenance of postharvest bitter melon fruit.

## Figures and Tables

**Figure 1 foods-09-01376-f001:**
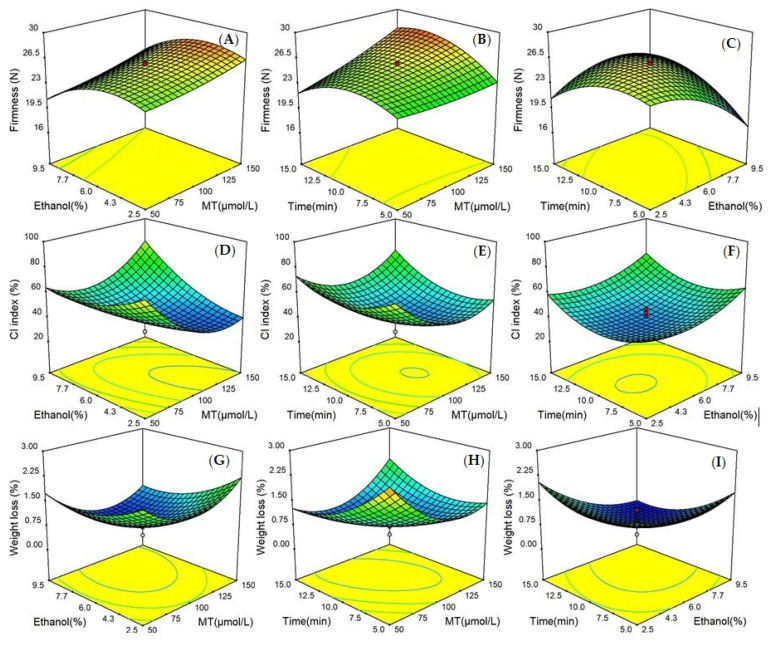
Response surface plots showed effects of treatment variables on firmness, CI index, and weight loss: (**A**,**D**,**G**) at varying MT and ethanol concentration; (**B**,**E**,**H**) at varying MT concentration and immersion time; (**C**,**F**,**I**) at varying ethanol concentration and immersion time.

**Figure 2 foods-09-01376-f002:**
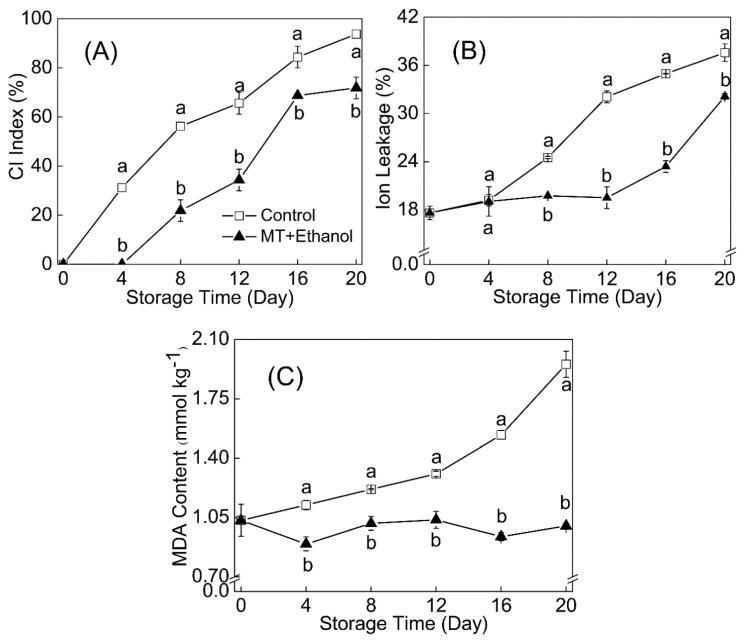
Effects of MT combined with ethanol treatment on CI index (**A**), ion leakage (**B**) and malondialdehyde (MDA) (**C**) content were presented. Results were expressed as mean ± SD of triplicate assays. Vertical bars represented the standard errors of the means. Significant differences (*p* < 0.05) were represented by different letters.

**Figure 3 foods-09-01376-f003:**
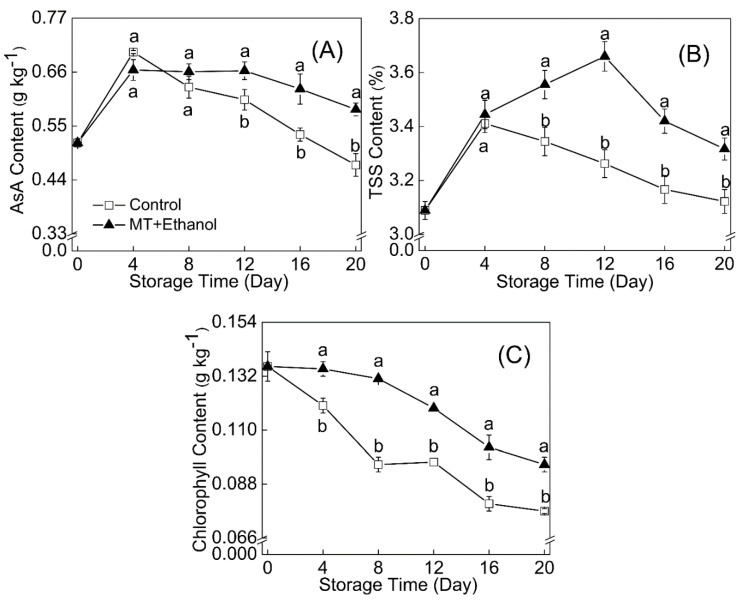
Effect of MT combined with ethanol treatment on ascorbic acid (AsA) (**A**), total soluble solids (TSSs) (**B**), and chlorophyll content (**C**) were presented. Results were expressed as mean ± SD of triplicate assays. Vertical bars represent the standard errors of the means. Significant differences (*p* < 0.05) are represented by different letters.

**Figure 4 foods-09-01376-f004:**
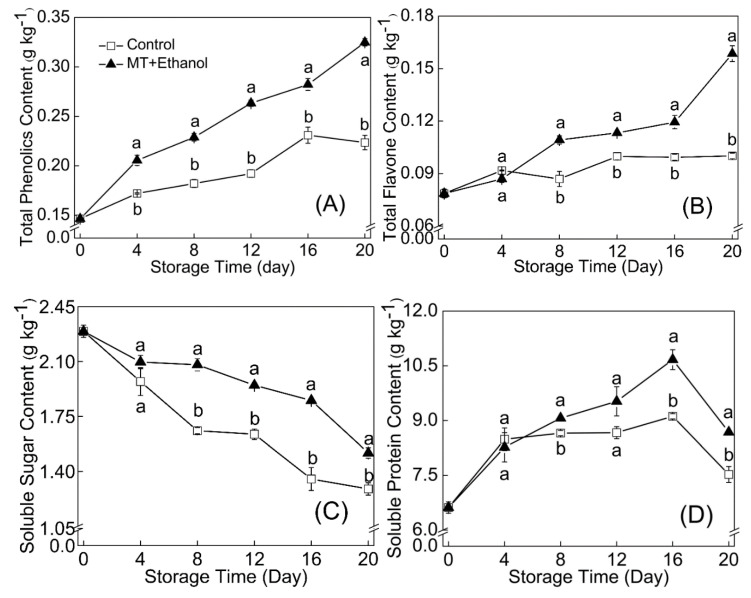
Effects of MT combined with ethanol treatment on total phenols (**A**), total flavonoids (**B**), soluble sugar (**C**), and soluble protein (**D**) contents. Results are expressed as mean ± SD of triplicate assays. Vertical bars represent the standard errors of the means. Significant differences (*p* < 0.05) are represented by different letters.

**Table 1 foods-09-01376-t001:** Coded and actual level of independent variables for the central composite design (CCD).

Run	x_1_ MT Concentration/µmol L^−1^	x_2_ Ethanol Concentration/%	x_3_ Treatment Time/min
1	50 (−1)	2.5 (−1)	5 (−1)
2	150 (1)	2.5 (−1)	5 (−1)
3	50 (−1)	9.5 (+1)	5 (−1)
4	150 (+1)	9.5 (+1)	5 (−1)
5	50 (−1)	2.5 (−1)	15 (+1)
6	150 (+1)	2.5 (−1)	15 (+1)
7	50 (−1)	9.5 (+1)	15 (+1)
8	150 (+1)	9.5 (+1)	15 (+1)
9	15.91 (−1.68)	6 (0)	10 (0)
10	184.09 (+1.68)	6 (0)	10 (0)
11	100 (0)	0.11 (−1.68)	10 (0)
12	100 (0)	11.89 (+1.68)	10 (0)
13	100 (0)	6 (0)	1.59 (−1.68)
14	100 (0)	6 (0)	18.41 (+1.68)
15	100 (0)	6 (0)	10 (0)
16	100 (0)	6 (0)	10 (0)
17	100 (0)	6 (0)	10 (0)
18	100 (0)	6 (0)	10 (0)
19	100 (0)	6 (0)	10 (0)
20	100 (0)	6 (0)	10 (0)

MT, melatonin.

**Table 2 foods-09-01376-t002:** Effect of melatonin (MT) and ethanol treatment alone on the chilling injury (CI) index of bitter melon fruit after storage at 4 °C for 20 d with a recovery period for 3 d at 25 °C.

Treatment	Concentration	CI Index/%
MT/µmol L^−1^	control	93.8 ± 12.5 ^a^
	50	87.5 ± 14.4 ^ab^
	100	62.5 ± 25.0 ^b^
	150	87.5 ± 14.4 ^ab^
	200	93.8 ± 12.5 ^a^
Ethanol/%	control	87.5 ± 25.0 ^a^
	3	87.5 ±14.4 ^a^
	6	50.0 ± 20.4 ^b^
	9	75.0 ± 20.4 ^ab^
	12	93.8 ± 12.5 ^a^
	15	93.8 ± 12.5 ^a^

Results are expressed as mean ± SD of triplicate assays. Different letters in a column imply statistically significant differences at the *p* < 0.05 level according to Duncan’s multiple range test.

**Table 3 foods-09-01376-t003:** Central-composite design with response values for firmness, CI index, and weight loss.

Run	Y_1_ Firmness/N		Y_2_ CI Index/%		Y_3_ Weight Loss/%	
	Actual Value	Predicted Value	Actual Value	Predicted Value	Actual Value	Predicted Value
1	25.35	25.35	90.75	94.85	2.98	3.10
2	25.28	24.87	37.50	42.10	2.17	2.03
3	17.17	16.96	78.25	78.85	3.30	3.30
4	17.42	17.44	72.00	83.98	1.47	1.60
5	18.70	18.68	97.00	90.82	2.44	2.24
6	23.52	23.72	56.25	61.45	3.33	3.26
7	20.11	20.52	72.00	73.19	0.99	1.06
8	26.52	26.52	100.00	101.70	1.64	1.45
9	24.33	24.22	97.00	99.97	3.02	3.00
10	28.75	28.86	90.75	79.59	2.31	2.43
11	21.87	22.01	56.25	54.46	2.36	2.50
12	17.45	17.31	81.25	74.85	1.19	1.15
13	18.35	18.71	78.25	68.40	2.49	2.39
14	21.10	20.74	78.25	79.91	1.34	1.54
15	24.82	25.56	47.00	40.40	0.79	0.77
16	25.46	25.56	43.75	40.40	0.72	0.77
17	25.92	25.56	28.25	40.40	1.22	0.77
18	25.72	25.56	40.75	40.40	0.71	0.77
19	25.71	25.56	43.75	40.40	0.75	0.77
20	25.71	25.56	37.50	40.40	0.46	0.77
Control	16.94		84.50		3.10	

**Table 4 foods-09-01376-t004:** ANOVA of each factor for the response surface quadratic model.

Coefficient Terms	Y_1_ Firmness/N		Y2 CI Index/%		Y_3_ Weight Loss/%	
	Regression Coefficients	*p*-Value	Regression Coefficients	*p*-Value	Regression Coefficients	*p*-Value
Constant Terms	25.56	<0.0001	40.40	0.0001	0.77	<0.0001
x1	1.38	<0.0001	−6.06	0.0272	−0.17	0.0224
x_2_	−1.40	<0.0001	6.06	0.0272	−0.40	<0.0001
x_3_	0.60	0.0002	3.42	0.1750	−0.25	0.0023
x_1_x_2_	0.24	0.1123	14.47	0.0008	−0.16	0.0817
x_1_x_3_	1.38	<0.0001	5.84	0.0856	0.52	<0.0001
x_2_x_3_	2.56	<0.0001	−0.41	0.8971	−0.35	0.0017
x_1_2	0.35	0.0067	17.46	<0.0001	0.69	<0.0001
x_2_2	−2.08	<0.0001	8.57	0.0037	0.37	0.0001
x_3_2	−2.06	<0.0001	11.93	0.0004	0.42	<0.0001
Lack of fit		0.5063		0.1810		0.6279
*R*2		0.9939		0.9270		0.9689
*R**Adj*2		0.9884		0.8613		0.9409

Level of significance: remarkably significant, *p*-value < 0.01; significant, *p*-value < 0.05; insignificant, *p*-value > 0.05.

**Table 5 foods-09-01376-t005:** Results of validation tests for CCD.

Group	MT/µmol L^−1^	Ethanol/%	Time/min	Firmness/N	CI index/%	Weight loss/%
MT–Ethanol	120	5.5	10	27.8 ± 0.5 ^a^	65.6 ± 4.4 ^c^	0.82 ± 0.011 ^c^
MT	100	-	10	27.3 ± 0.2 ^a^	84.4 ± 4.4 ^b^	0.86 ± 0.023 ^b^
Ethanol	-	6	10	26.6 ± 0.1 ^b^	87.5 ± 0.0 ^a^	0.85 ± 0.001 ^b^
Control	-	-	10	25.2 ± 0.3 ^c^	90.6 ± 4.4 ^a^	0.93 ± 0.015 ^a^

Results are expressed as mean ± SD of triplicate assays. Different letters in a column imply statistically significant differences at the *p* < 0.05 level according to Duncan’s multiple range test.

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
