# Peer review of "A Combination of Melatonin and Ethanol Treatment Improves Postharvest Quality in Bitter Melon Fruit"

_foods, 2020, doi:10.3390/foods9101376_

Round 1

Reviewer 1 Report

The manuscript entitled "A combination of melatonin and ethanol treatment improves postharvest quality in bitter melon" approaches the interesting subject of hormones application as postharvest technology for quality preservation. The subject is innovative but the manuscript needs major language revisions. If corrected, the manuscript represents an interesting and innovative addition for postharvest quality research.However, attention is needed on the use of correct English to avoid incorrections and possible confusing statements. 

Another point to raise is the fact that the Abstract is basically equal to the Conclusions section. Improvement is needed. 

The Materials and Methods section needs some detail on how assays were performed or adjusted to the referenced methods. Not extensively but brief descriptions of the referenced methods.

Also, no test was performed to understand the physiological effect of melatonin on bitter melon postharvest quality. The manuscript shows and states the positive effect but does not discuss it at the physiological level. No explanation or hypothesis is raisen. For instance, why the only significant decreasing effect of 100 umoL/L or 6% ethanol on CI Index?  This aspect needs discussion. 

Author Response

Response to Reviewer 1 Comments

Point 1: The manuscript entitled "A combination of melatonin and ethanol treatment improves postharvest quality in bitter melon" approaches the interesting subject of hormones application as postharvest technology for quality preservation. The subject is innovative but the manuscript needs major language revisions. If corrected, the manuscript represents an interesting and innovative addition for postharvest quality research. However, attention is needed on the use of correct English to avoid incorrections and possible confusing statements. 

Response 1: Thank you for your kind comments. The language quality of the original manuscript of our paper is indeed insufficient. We've send the manuscript to a native English speaker for grammer editing.

Point 2: Another point to raise is the fact that the Abstract is basically equal to the Conclusions section. Improvement is needed.

Response 2: We totally agree with the viewer, and have revised the Conclusions section to address your concerns. Please see page 14 of the revised manuscript, lines 405-410.

Point 3: The materials and methods section need some detail on how assays were performed or adjusted to the referenced methods. Not extensively but brief descriptions of the referenced methods.

Response 3: We have corrected the materials and methods section according to the reviewer’s comments and here we did not list the changes but marked in red in revised paper (page 4 to 5).

Point 4: Also, no test was performed to understand the physiological effect of melatonin on bitter melon postharvest quality. The manuscript shows and states the positive effect but does not discuss it at the physiological level. No explanation or hypothesis is raisen. For instance, why the only significant decreasing effect of 100 μmoL L-1 or 6 % ethanol on CI Index?

Response 4: We totally understand the reviewer’s concern. We studied the physiological effects of melatonin on postharvest bitter melon during cold storage, while we plan to present the results in another paper. Besides, in this paper, we focus on the effects of treatment on the quality of bitter melon fruit, rather than physiological aspects. Besides, some arguments were also added to discuss the positive physiological effect of melatonin on bitter melon, lines 360-364 on page 13. In addition, we have added some description in the revised manuscript (page 12, lines 340-344) “When MT concentration increased to 200 µmol L-1 or ethanol concentration increased to 12 % and 15 %, serious CI symptoms would appear. It is probably that higher levels of MT or ethanol may cause the chemical injury to the cell of bitter melon fruit. Meanwhile, the osmotic stress caused by high osmotic pressure may arise, which will result in a negative effect of MT and ethanol treatment”. Moreover, CI index of 150 µmol L-1 melatonin or 9 % ethanol treated fruit was slightly lower than the control with no significant difference (p > 0.05). We only used the treatment of 100 μmoL L-1 melatonin or 6 % ethanol which had significant difference (p < 0.05) with control for the following experiment.

Reviewer 2 Report

Manuscript review: foods-923969
Title: A combination of melatonin and ethanol treatment improves postharvest quality in bitter melon fruit

The manuscript reports results from the application of a postharvest treatment to prevent chilling injury (CI) in bitter melon fruit. According to the authors, immersion of bitter melon in a solution of melatonin plus ethanol for 10 minutes alleviates symptoms of CI after storage for 20 days at 4C. Overall, the experiments are well designed and the results sound. Following are some suggestions to improve the quality of the manuscript:

• Polish the English: some sentences are too long and hard to understand, particularly in the
Discussion section.
• In the introduction explain better the importance of the study; chilling injury in bitter melon;
advantages of using CCD (explain better the theory behind and practical applications in
postharvest); explain better what is RSM and why did you use I; practical applications of the
melatonin + ethanol treatment.
• Lines 78-87: why was ethanol added to the melatonin solution in a single factor experiment with
melatonin? Why was ethanol also added to distilled water (control)? Why was soaking
conducted under dark conditions? Why did you use soaking?
• Lines 95-110: this section needs to be better explained. The data contained on the table needs
to be better explained as well as the purpose of conducting RSM.
• Explain why you used a 3 days storage at 25C after exposure for 20 days at 4C.
• Figures can be rounded: for example, 33.33% can be rounded to 33.3% or 33%.
• Discuss why concentrations of melatonin higher than 100 do seem to prevent CI in bitter melon.
What are the physiological implications of using higher levels of melatonin?
• Explain the purpose of using model fitting?
• Is total phenol and flavonoids accumulation due to synthesis or to concentration due to water
loss during storage? Try convert to dry weight and verify if the increase is not due to
accumulation rather than to synthesis.

Author Response

Response to Reviewer 2 Comments

Point 1: Polish the English: some sentences are too long and hard to understand, particularly in the discussion section.

Response 1: We feel sorry for the inconvenience brought to the reviewer. We have made correction according to the reviewer’s comments and hope that it will meet with approval. We did not list the modified text but marked in red in revised paper.

Point 2: In the introduction explain better the importance of the study; chilling injury in bitter melon; advantages of using CCD (explain better the theory behind and practical applications in postharvest); explain better what is RSM and why did you use I; practical applications of the melatonin + ethanol treatment.

Response 2: We appreciate for your valuable suggestion. As requested by the reviewer, we have added a paragraph in the revised manuscript (page 2, lines 57-63). However, to the best of our knowledge, little information concerning the combined application of melatonin and ethanol on fruit preservation was reported. We’re sorry that we can't list the practical applications related to melatonin-ethanol combined treatment.

Point 3: Lines 78-87: why was ethanol added to the melatonin solution in a single factor experiment with melatonin? Why was ethanol also added to distilled water (control)? Why was soaking conducted under dark conditions? Why did you use soaking?

Response 3: We have mentioned the reasons in the revised manuscript (lines 75-76 on page 2) “As melatonin was light sensitive, the experiment was carried out in dark conditions. Besides, melatonin was insoluble in water, and 0.0625% ethanol was used as co-solvent to promote the dissolution of melatonin.”. In addition, we referred to the previous studies on the application of melatonin and ethanol in postharvest fruit, and choose the soaking method. Melatonin was usually used to treat fruit by soaking, while ethanol had been reported on both soaking and fumigation. Furthermore, in view of the better penetration effect of immersion treatment and the convenience of experimental operation, the soaking method was chosen eventually.

Point 4: Lines 95-110: this section needs to be better explained. The data contained on the table needs to be better explained as well as the purpose of conducting RSM.

Response 4: We agree with the viewer, and have revised the text to address your concerns and hope that it is now clearer. Please see page 3 of the revised manuscript, lines 93-99 and 103-104.

Point 5: Explain why you used a 3 days storage at 25 ℃ after exposure for 20 days at 4 ℃.

Response 5: Generally, chilling injury symptom could be shown after a few days at room temperature. For bitter melon fruit in our study, dark watery patches and black surface discoloration as chilling injury symptom were shown after removing from 4 ℃ into 25 ℃ for 3 days. We used a three days storage at 25 ℃ after exposure for 20 days at 4 ℃ to manifest the symptoms of chilling injury more obviously.

Point 6: Figures can be rounded: for example, 33.33% can be rounded to 33.3% or 33%.

Response 6: We appreciate for your valuable comment. As suggested by the reviewer, we have rounded these figures and marked in red in revised paper (page 5-12, lines 194-328).

Point 7: Discuss why concentrations of melatonin higher than 100 do seem to prevent CI in bitter melon. What are the physiological implications of using higher levels of melatonin?

Response 7: We have added some descriptions to explain the physiological implications of using higher levels of melatonin. Please see page 12 of the revised manuscript, lines 340-344. CI index of 150 µmol L-1 melatonin treated fruit was slightly lower than the control, while it had no significant difference (p > 0.05) in comparison with control. The treatment of 150 µmoL L-1 melatonin may not cause the chemical injury or osmotic stress to the cell of bitter melon fruit. Thus, concentrations of melatonin higher than 100 µmol L-1 seemed to prevent CI in bitter melon.

Point 8: Explain the purpose of using model fitting?

Response 8: It was necessary to fit a mathematical equation according to the level of the studied values to describe the response behavior after obtaining the data related to each experimental point. However, the regression equation was not enough to show the fitting effect, then ANOVA was used to evaluate the fitting quality of the model more reliably. Therefore, we used model fitting of regression equation and ANOVA to show the effect of response.

Point 9: Is total phenol and flavonoids accumulation due to synthesis or to concentration due to water loss during storage? Try convert to dry weight and verify if the increase is not due to accumulation rather than to synthesis.

Response 9: The water loss of bitter melon during storage was very little, so we think that the accumulation of total phenols and flavonoids may not be related to water loss. In addition, as the important antioxidants in plant, the increasing of them was a response to chilling stress. Previous studies had demonstrated that total phenols and flavonoids content can be greatly promoted by melatonin or ethanol treatment. Our current study also proved that. Thus, the accumulations of phenol and flavonoids in bitter melon during storage were probably due to the synthesis of them.